# Long Intergenic Non-Coding RNAs of Human Chromosome 18: Focus on Cancers

**DOI:** 10.3390/biomedicines12030544

**Published:** 2024-02-28

**Authors:** Pavel V. Ershov, Evgeniy O. Yablokov, Yuri V. Mezentsev, Alexis S. Ivanov

**Affiliations:** Institute of Biomedical Chemistry, Moscow 119121, Russia; evgeniy.yablokov@ibmc.msk.ru (E.O.Y.); yu.mezentsev@gmail.com (Y.V.M.); alexei.ivanov@ibmc.msk.ru (A.S.I.)

**Keywords:** chromosome 18, open-reading frame, cancers, non-coding RNA, long-intergenic non-coding RNA, RNA–RNA interactions, RNA–protein interactions, transcriptomic signatures

## Abstract

Malignant neoplasms are characterized by high molecular heterogeneity due to multilevel deregulation of gene expression and cellular functions. It is known that non-coding RNAs, including long intergenic non-coding RNAs (lincRNAs), can play significant roles in cancer biology. The current review focuses on a systematical analysis of genomic, transcriptomic, epigenomic, interactomic, and literature data on 65 lincRNAs of human chromosome 18 in the context of pan-cancer studies. The entire group of lincRNAs can be conditionally divided into 4 subgroups depending on experimental evidence on direct or indirect involvement in cancers and the biological associations with cancers, which we found during the data-mining process: the most studied (5 lincRNAs), moderately or poorly studied (11 lincRNAs), and understudied (31 lincRNAs). For the remaining 18 lincRNAs, data for analysis were fragmentary or missing. Among the key findings were the following: Of the lincRNAs of human chromosome 18, 40% have tissue-specific expression patterns, 22% of lincRNAs are known to have gene fusions, 40% of lincRNAs are prone to gene amplifications and/or deletions in cancers at a frequency greater than 3%, and 23% of lincRNAs are differentially expressed across cancer types, whereas 7% have subtype-specific expression patterns. LincRNAs’ interactomes consist of ‘master’ microRNAs and 47 proteins (including cancer-associated proteins and microRNAs) that can interact with 3 or more lincRNAs. Functional enrichment analysis of a set of highly co-expressed genes retrieved for 17 lincRNAs in different cancer types indicated the potential associations of these lincRNAs with cellular signaling pathways. Six lincRNAs encoded small open-reading frame (smORF) proteins with emerging roles in cancers, and microRNAs as well as proteins with known functions in molecular carcinogenesis can bind to coding regions of smORFs. We identified seven transcriptomic signatures with potential prognostic value, consisting of two to seven different lincRNAs only. Taken together, the literature, biomedical, and molecular biology data analyzed indicated that only five of all lincRNAs of human chromosome 18 are cancer-associated, while eleven other lincRNAs have the tendency to be associated with cancers.

## 1. Introduction

Long non-coding RNAs (lncRNAs) have been arbitrarily defined as non-coding transcripts of more than 200 nucleotides (200 nt) [1], but today, the term ‘lncRNA’ defines a more expanded list of different RNAs [2]. With respect to protein-coding genes, lncRNAs can be intergenic, antisense, or intronic. Long intergenic non-coding RNAs (lincRNAs) are autonomously transcribed RNAs whose genes do not trespass on nearby protein-coding loci [2,3]. It is believed that lincRNAs are expressed in a tissue-, cell-, stage-, and disease-specific manner. There are several known universal mechanisms by which all lncRNAs, and lincRNAs in particular, realize their biologic functions, such as control of chromatin architecture, modulation of enhancer activity, formation of biomolecular condensates [2], and epigenetic and transcriptional regulation of gene expression [3]. LincRNAs can influence other biomolecules, acting as signals, decoys, scaffolds, and guides [4] that are mediated by binding with chromatin and chromatin-modifying complexes [5,6], transcription factors [7], RNA-binding proteins (RBPs) [8], and various types of non-coding RNAs [4,9].

It is known that lincRNAs affect biological pathways in autoimmune and neurodegenerative disorders [10], cardiovascular diseases [9], inflammation [11,12], and normal and malignant hematopoiesis [13]. LincRNAs are aberrantly expressed in various malignant tumors [14]. For example, aberrant expression of *LINC00173* affects the initiation and progression of human cancers [15], while *LINC01094* indirectly stabilizes the brain-derived neurotrophic factor via microRNA miR-577 in glioblastoma cells [16]. Overexpression of *LINC01355* significantly inhibits tumorigenesis of breast cancer cells through interaction and stabilization of forkhead box O3 protein (FOXO3), leading to transcriptional repression of the cyclin D1 gene [15]. *LINC00680* enhances hepatocellular carcinoma stem cell behavior and chemoresistance by sponging miR-568 to upregulate AKT Ser/Thr protein kinase 3 [17]. Thus, some lincRNAs involved in the pathogenesis of various cancer or non-cancer diseases may be considered as potential molecular targets or prognostic, predictive, and diagnostic biomarkers. Recent studies of human chromosome 18 within the framework of the Russian segment of the international program ‘The Human Proteome Project’ have investigated a detailed proteogenomic landscape of human chromosome 18 genes in the HepG2 cell line and liver tissue [18,19,20]. Taken together, literature data speak in favor of the associations between some lincRNAs of human chromosome 18 and cancers. The goal of the current review is a systematic analysis of genomic, transcriptomic, epigenomic, interactomic, and literature data on 65 lincRNAs of human chromosome 18 in the context of pan-cancer studies.

## 2. A Spectrum of Genes Encoding lincRNAs of Human Chromosome 18

Sixty-five genes of human chromosome 18 encode lincRNAs-of-interest (Appendix A). The lengths of lincRNAs range from 297 to 6201 nucleotides. LincRNAs’ transcripts per gene vary from 1 to 84 due to the alternative splicing. Records on subcellular localization in the RNAlocate database [21] are available only for 22 of 65 lincRNAs. They are localized in the circulating blood exosomes, nucleus, nucleoplasm, membranes, and cytosol. Four linc-genes (*LINC00305*, *LINC00470*, *LINC00526*, and *LINC01387*) encode open-reading frames of uncharacterized chromosome-specific proteins C18orf20, C18orf2, C18orf18, and C18orf64, respectively. However, their protein existence status remains ‘uncertain’ according to the PepPsy portal [22]. LincRNAs-of-interest in the form of circRNAs, a type of single-stranded RNA forming a covalently closed continuous loop, were not found in the ‘Circular RNA Interactome’ [23] and ‘CircBank’ [24] databases. Knockouts of *LINC01387*, *LINC01899*, and *LINC01909* genes resulted in significant biological effects in different cell lines (≥four independent sources, according to the BioGRID Open Repository of CRISPR Screens v.1.1.14 [25]). At least 8 lincRNAs are associated with 17 different types of malignancies (Figure 1), which follows from the RNA-Disease Repository v. 4.0 database [26].

### 2.1. Alterations of Genes Encoding lincRNAs of Human Chromosome 18

Single-nucleotide polymorphism (SNP) of linc-genes can lead to the disruption of gene function or production of defective RNAs with altered secondary structures. SNPs are described for 41 of 65 lincRNAs-of-interest (Appendix A). For example, 108, 234, and 259 pathogenic variants are known for *LINC01477*, *LINC01917*, and *LINC02565* genes, respectively (Appendix A).

Gene fusions due to the inter- or intra-chromosomal rearrangements are the common gene alterations in tumor cells. They ultimately affect the expression of chimeric RNAs encoding protein products with abnormal activity. Another source of chimeric RNAs is the alternative intergenic splicing [27,28]. 

The landscape of fusions with participation of linc-genes of human chromosome 18 in cancer cell lines was explored using the ‘Cancer Dependency Map’ database [29]. Slightly more than half (63%) of the found gene fusions are within the same loci as a linc-gene or nearby chromosome loci (Appendix A). Fusions of linc-genes and cancer-associated genes, such as *RAD54L* (RAD54 like), *PIK3C3* (phosphatidylinositol 3-kinase catalytic subunit type 3), *MBP* (myelin basic protein), *PFN2* (profilin 2), *MBD2* (methyl-CpG binding domain protein 2), and *YES1* (YES proto-oncogene 1, Src-family tyrosine kinase), as well as tumor suppressors *SMAD4* (SMAD family member 4) and *SDHA* (succinate dehydrogenase complex flavoprotein subunit A), can be an additional factor in cancer promotion.

The landscape of amplifications and deletions of linc-genes of human chromosome 18 was examined in 11 different cancer types (with >100 cases in each cohort) using the cBioPortal database [30] (Appendix A). There are several conventional subgroups of linc-genes with frequency of deletions >3%—D1 (*LINC00305* and *LINC00907* in metastatic breast cancer (mBRCA), esophageal carcinoma (ESCA), and pancreatic adenocarcinoma (PAAD) and D2 (*LINC01538*, *LINC01541*, *LINC01544*, *LINC01924*, *LINC02582*, and *LINC02864* in ESCA, head and neck cancer (HNSC), and PAAD), as well as amplifications > 3%—A1 (*LINC00470*, *LINC00526*, *LINC00667*, and *LINC00668* in bladder cancer (BLCA), mBRCA, ESCA, PAAD, prostate adenocarcinoma (PRAD), and stomach adenocarcinoma (STAD)) and A2 (*LINC01387*, *LINC01543*, and *LINC01915* in ovarian cancer (OV), PAAD, PRAD, and STAD). The highest frequency of deletions (16–25%) is observed in PAAD and mBRCA, while the highest frequency of amplifications—9.8% and 8.7%—is observed in PRAD (*LINC00907*) and PAAD (*LINC01915*), respectively.

### 2.2. Promoter Methylation of Genes Encoding lincRNAs of Human Chromosome 18

Differential promoter methylation patterns are found in The Cancer Genome Atlas (TCGA) pan-cancer cohort (*n* > 30 cases in each cohort) using the web-based tool DNMIVD [31] for *LINC00470* in colorectal adenocarcinoma (COAD; Appendix A), for *LINC00305* in head and neck cancer (HNSC), liver hepatocellular carcinoma (LIHC), lung adenocarcinoma (LUAD), and lung squamous cell carcinoma (LUSC; Appendix A), and for *LINC00526* in uterine corpus endometrial carcinoma (UCEC; Appendix A). At the same time, there are associations of *LINC00526* gene promoter methylation with patients’ survival rates in UCEC (46 normal and 432 tumor cases; Figure 2A) and an inverse relationship between an increase in promoter methylation of *LINC00526* gene and a decrease in its gene expression levels (Figure 2B). 

### 2.3. Differentially Expressed Genes Encoding lincRNAs of Human Chromosome 18

A landscape of tissue-specific expression of 47 of 65 lincRNAs of human chromosome 18 was analyzed using the Gene Tissue Expression Portal (GTEx portal, https://www.gtexportal.org, accessed on 5 February 2024; Appendix A). Two lincRNAs, encoded by *LINC00526* and *LINC00667* genes, are expressed in almost all tissues examined >3 TPM (Transcript per Million). It is also interesting to note that 14 different lincRNAs have tissue-specific expression in testis tissue (>3 TPM). Other lincRNAs with expression levels > 3 TPM are as follows: *LINC01544* (brain cerebellum and cerebellar hemisphere), *LINC01909* (liver tissue), *LINC00668* (colon transverse and testis tissues), *LINC01539* (thyroid and testis tissues), *LINC01543* (kidney medulla), *LINC01926* (prostate and testis tissues), and *LINC01444* (thyroid tissue). Thus, it can be seen that the tissue-specific expression patterns are characteristic for almost 30 lincRNAs.

Further, eight differentially expressed linc-genes in cancers were selected using the web-based tool GEPIA2 [32] at |log_2_fold-change (FC)| > 1.5 tumor/normal tissue and *p*-value < 0.01 using TCGA as a data source, without taking into account cancer subtypes. All the boxplots, depicting differentially expressed linc-genes of human chromosome 18 in different cancer types, are presented in Appendix A. Of them, downregulation of *LINC00305* (Appendix A), *LINC00470* (Appendix A), *LINC00668* (Appendix A), *LINC01255* (Appendix A), *LINC01478* (Appendix A), and *LINC01539* (Appendix A) genes occurs in testicular germ cell tumors (TGCT), of *LINC00470* (Appendix A), *LINC00526* (Appendix A), and *LINC00668* (Appendix A) genes occurs in acute myeloid leukemia (LAML), and of *LINC01539* and *LINC00667* genes occurs in thyroid cancer (THCA; Appendix A) and UCEC (Appendix A), respectively. Upregulation of *LINC00668* gene occurs in COAD, READ, STAD, and LUSC (Appendix A), of *LINC00526* and *LINC00667* genes occurs in thymoma (THYM; Appendix A), and of *LINC01443* gene occurs in skin cutaneous melanoma (SKCM; Appendix A).

Next, cancer subtype-specific expression patterns of linc-genes were examined using the web-based tool GEPIA2 at |log_2_FC| > 1 tumor/normal tissue and *p*-value < 0.05 (Appendix A). Subtype-specific differential expression of six genes is found in eight cancer types. Statistically significant induction of gene expression occurs for the following lincRNAs: *LINC00470* in basal and classical subtypes of lung squamous carcinoma (Appendix A), *LINC00526* in the pro-neural subtype of glioblastoma (Appendix A), and *LINC01415* in oligodendroglioma (a subtype of low-grade glioma). The downregulation of gene expression is characteristic for *LINC00667* in the papillary subtype of bladder cancer (Appendix A), *LINC00667* in basal-like and HER2 breast cancer (Appendix A), and *LINC00667* in colorectal adenocarcinoma with high microsatellite instability (Appendix A). *LINC00668* is upregulated in microsatellite-stable colorectal and rectal adenocarcinoma with low microsatellite instability (Appendix A, respectively). Changes in *LINC00470* gene expression occur between classical and primitive subtypes in lung squamous carcinoma (Appendix A). *LINC00668* gene expression varies in rectal adenocarcinoma between subtypes with high and low microsatellite instability (Appendix A). There are no subtype-specific gene expression changes between seminoma and non-seminoma subtypes of six linc-genes (Appendix A), with differential expression in TGCT as well as between subtypes of skin cutaneous melanoma in the case of *LINC01443* (Appendix A).

Statistically significant changes of stage-specific expression of linc-genes in cancers were not found (F-test, *p*-value ≤ 0.05).

A list of the top-100 highly co-expressed genes (r ≥ 0.8) for the above-mentioned differentially expressed linc-genes of human chromosome 18 in cancer types was retrieved from TCGA and GTEx portals using the web-based tool GEPIA2. The list, containing co-expressed genes and results of functional enrichment analysis, is presented in Appendix A. It is shown that *LINC00668*, *LINC01478*, and *LINC01539* genes are co-expressed with genes participating in the meiotic cell cycle and cellular process involved in reproduction in multicellular organisms and motile cilium in testis tissue. *LINC00526* and *LINC00667* genes are co-expressed with genes participating in the processing of capped intron-containing pre-mRNA, RNA processing, and transcriptional and post-transcriptional regulation of gene expression in thymoma.

Analysis of the NCBI Gene Expression Omnibus (GEO; ≥10 cases in each dataset) on condition-specific expression of linc-genes of human chromosome 18 in tumor tissues or cell lines was performed. We analyzed 36 relevant datasets (Appendix A) that were previously selected by GEO Profiles to search for differently expressed linc-genes using the GEO2R tool at |log_2_FC| ≥ 0.8 and *p*-value < 0.05. Ten differently expressed linc-genes were found (Appendix A) with expression patterns specific to certain cancer types, metastasis, and therapy effects [33,34,35,36].

### 2.4. Transcriptional Regulation of Genes Encoding lincRNAs of Human Chromosome 18

LincRNAs’ accumulation in tumor tissues may be related to transcriptional regulation via different combinations of transcriptional factors (TFs). We analyzed data on 350 potential TFs, whose binding sites were predicted in the promoter or enhancer regions of 30 linc-genes (Appendix A) using the GeneHancer Regulatory Elements in the frame of the GeneCards database [37]. All findings on potential TFs found for genes, encoding lincRNAs of human chromosome 18, are shown in Appendix A. Non-overlapping groups I and II include TFs, each of which interacts with DNA regions of >15 and >10 linc-genes, respectively (Appendix A). Some genes, encoding TFs, were upregulated in cancers, but functional enrichment analysis of TF sets did not show their over-representation in cancer-associated pathways from both groups. It should be noted that the aspects of transcriptional regulation of genes encoding lincRNAs of human chromosome 18 are practically not studied.

## 3. Interactomics of lincRNAs of Human Chromosome 18

### 3.1. Interactions of lincRNAs with microRNAs

The post-transcriptional regulation can be realized via interactions of lincRNAs with microRNAs. The miRDB database [38] was used to predict a number of interacting microRNAs based on the lincRNAs’ sequences. As a result, 1004 microRNA/lincRNA interactions were found (Appendix A). Of them, 57% of microRNAs interact with only one lincRNA, 27.5% with 2 different lincRNAs, 11% with 3 lincRNAs, and 4% with 4 lincRNAs. One microRNA, hsa-miR-670-3p, was predicted to interact with five different lincRNAs. Figure 3 shows a core part of a microRNAs/lincRNAs subnetwork, from which it follows that the highest connectivity was observed for *LINC02864* (degree = 9), *LINC01902* (degree = 7), and *LINC03035* (degree = 6). Of 1004 predicted interactions, 33 have been previously verified in experiments recorded in the LncBook v. 2.0 database [39] (Appendix A).

The RNA–RNA interaction network is quite complex to find disease-specific associations, and the number of predicted hypotheses may be excessive, despite the stringent selection criteria. We performed a functional enrichment analysis with the Kyoto Encyclopedia of Genes and Genomes (KEGG) and Gene Ontology terms of experimentally verified microRNA/lincRNA interactions with each of six lincRNAs of human chromosome 18 (Appendix A) using the web-based tool miEAA v.2.1 [40].

The potential for participation in cancer-associated processes decreases in a set of *LINC00667*, *LINC00668* > *LINC00470* > *LINC01926*, *LINC01544*, *LINC01909*, depending on the number of functional terms. Further, using the PlasmiR database [41], we additionally searched records on the diagnostic value of 17 microRNAs interacting with these lincRNAs (excluding *LINC01909*) in plasma and serum blood samples (Appendix A). Since lincRNAs act as molecular sponges for microRNAs [42], tissue-specific accumulation of lincRNAs may indirectly influence microRNA content in tissues and, hence, secretion to the biological fluids.

Generally speaking, microRNAs are capable of regulating the half-life of lncRNAs. The latter affects molecular processes and biological functions, and changes in microRNAs’ content directly alter the cellular responses in pathological conditions [43]. However, a reverse phenomenon is also known. Thus, a high negative correlation between the levels of lncRNA OIP5-AS1 and miR-7 was associated with OIP5-AS1-mediated miR-7 degradation, which promoted myotube formation by stimulating a myogenic fusion program [44]. Opposite expression directions of three linc-genes and interacting microRNAs (Table 1) may imply the existence of cancer-dependent regulation of microRNA transcripts’ half-life. For example, *LINC00668* and miR-236-3p are up- and down-regulated in READ, respectively. Otherwise, *LINC01539* and miR-34a-5p are down- and up-regulated in THCA, respectively.

### 3.2. Interactions of lincRNAs of Human Chromosome 18 with Other lncRNAs

The analysis of the landscape of co-expressed genes allows us to predict the functional processes and molecular pathways, with which lincRNAs of human chromosome 18 can be associated under normal and disease conditions, in particular, cancer. For this, lists of the top-100 co-expressed genes for each of eleven differentially expressed lincRNAs in a set of cancer types were retrieved from TCGA database using the web-based tool GEPIA2, with a Pearson correlation coefficient ≥ 0.7 as a cut-off (Appendix A). We also analyzed lists of highly co-expressed genes retrieved regarding lincRNAs without differential expression in cancer types. Since plenty of co-expressed genes represented non-coding RNAs and pseudogenes, we used the NCpath web-based tool [46] adapted for functional enrichment analysis of gene sets containing non-coding RNAs. All the pathway terms are shown in Appendix A, and the most frequent pathway terms are summarized in Appendix A. It shows that some lincRNAs of human chromosome 18 may be associated with signaling pathways in cancers under conditions of probable interactions with other non-coding RNAs having similar expression patterns. Thus, *LINC00526* and *LINC00667* are represented in the majority of pathways in different cancer types. The more cancer-specific lincRNAs are *LINC00470*, *LINC00668*, *LINC00907*, *LINC01254*, *LINC01415*, *LINC01478*, and *LINC01539*. Finally, there is a subset of *LINC01378*, *LINC01443*, *LINC01477*, and *LINC01544*, which are typically represented in one or two pathways in only one cancer type. The most common pathways, with which lincRNAs of human chromosome 18 are associated, are ‘adherent junction’, ‘focal adhesion’, ‘FoxO’, and ‘mTOR’.

### 3.3. Interactions of lincRNAs of Human Chromosome 18 with Cellular Proteins

A list of 613 binary interactions between lincRNAs of human chromosome 18 and cellular proteins was retrieved from four different databases (LncTarD v. 2.0 [47], RNAinter [48], NPInter v.5.0 [49], and Biogrid v.4.4 [25]; Appendix A). Further, 47 proteins, which bind with at least ≥3 lincRNAs, were selected (Appendix A). Of them, histones H3 methylated at lysine 4 or lysine 27 or acetylated at lysine 27 are the most common interactors for almost all 46 lincRNAs (Appendix A). A functional enrichment analysis and protein–protein interaction analysis of 42 non-histone proteins using the tool WebGestalt [50] showed that lincRNA-binding proteins are involved in the epigenetic regulation of gene expression and mRNA processing (Table 2).

There are also a number of proteins: AR, ESR1, EWSR1, FOXA1, IGF2BP3, HNF4A, POU5F1, SMARCA4, and SOX2, that are associated with epithelial cancers, urogenital neoplasms, and adenocarcinomas, and capable of forming a highly connected subnetwork of protein–protein interactions (Appendix A). In addition, genes encoding these proteins, excluding *IGF2BP3* and *HNF4A*, are causally implicated in cancer promotion [51] due to increased rates of driver mutations.

Generally, the direct interactions of lncRNAs with histone proteins, and especially TFs, are a type of transcriptional regulation through activation or recruitment of TFs [52,53], and these events may have a cancer-specific pattern [54]. Moreover, in the LncBase v.3.0 database [55], we found that experimentally supported interactions of a number of genes: *LINC00470*, *LINC00526*, *LINC00667*, *LINC00668*, and *LINC02582*, with cellular proteins are associated with increased growth, proliferation, migration, and invasion of tumor cells, as well as tumor progression through apoptosis suppression and a decrease in radio-sensitivity (Appendix A). An interesting fact is that TF FOXA1 (forkhead box protein A1) interacts with 21 lincRNAs-of-interest, while *LINC00907*, *LINC01255*, and *LINC01910* gene expressions are predicted to be regulated by FOXA1 (GeneCards database). The same coincidence was observed in three other cases: *LINC00526* and *LINC01910* gene expression may be regulated by HNF4A (hepatocyte nuclear factor 4α), *LINC01919* by POU5F1 (POU domain, class 5, transcription factor 1), and *LINC00470* by SMARCA4 (SWI/SNF-related, matrix-associated, actin-dependent regulator of chromatin, subfamily a, member 4). 

### 3.4. Small Open-Reading Frame Proteins (smORF-Proteins)

As a rule, lincRNAs have no protein-coding potential, but some of them are annotated as protein-coding small open-reading frames (smORFs) and thus can be considered as coding RNAs. Eukaryotic proteins encoded by such RNAs are known as ‘small proteins’ that are usually from 15 to 100 amino acid residues in length [56,57]. As an example, a small protein derived from *LINC00675* was detected only in tumor tissues [58], and participates in cancer-associated processes [58]. The protein-coding or peptide-coding potential of lincRNAs of human chromosome 18 was examined using the LncBook 2.0 database linked with the SmProt database [59]. SmProt is an online repository with annotation of small proteins derived from ribosome profiling that is based on high-throughput sequencing of mRNAs interacting with active ribosomes in cells [60]. Appendix A shows a list of 28 potential smORF proteins encoded by six lincRNAs of chromosome 18, though records on their mass spectrometry identification are absent in the Peptide Atlas database [61]. 

Theoretically, interference of RNAs’ open-reading frames, while interacting with other types of RNAs or RNA-binding proteins (RBPs), modulates the translation of small proteins during the neoplastic transformation of cells. microRNAs and RBPs, interacting with lincRNA sequences, encoding smORF proteins, were predicted using the miRDB [38] and RBPmap [62] tools, respectively. Appendix A shows that predicted interactors of lincRNAs include both cancer-associated microRNAs and RBPs. As it can be seen, RBPs (SRSF2 and SRSF10 (serine- and arginine-rich splicing factor 2 and 10), RBM4 and RBM25 (RNA-binding motif protein 4 and 25), MBNL1 (muscle-blind-like splicing regulator 1), CNOT4 (CCR4-NOT transcription complex subunit 4), TRA2A (transformer 2 alpha homolog), and HNRNPs (heterogeneous nuclear ribonucleoproteins) may act as potential interactors for multiple lincRNAs of chromosome 18. The largest number of interactors (3 microRNAs and 17 RNA-binding proteins) was predicted for SPROHSA300118, which is one of the smORF proteins encoded by *LINC00667*.

## 4. Prognostic and Predictive Value of Genes Encoding lincRNAs of Human Chromosome 18

Transcriptomic signatures with participation of non-coding RNAs, in particular lincRNAs, can be associated with disease prognosis or prediction of therapy responses. In this regard, Nie and co-authors [63] found that high tissue expression levels of *MNX1-AS1*, *LINC00330*, and *LSAMP-AS1* genes in laryngeal cancer correlated with low survival rates, and the high-risk group was sensitive to AKT (protein kinase B) inhibitors. Hence, we searched for associations between gene expression levels of lincRNAs of human chromosome 18 and survival rates of patients with cancer using the Kaplan–Meier plotter [64,65]. All findings regarding lincRNAs of human chromosome 18 are shown in Appendix A. Several transcriptomic signatures, containing from four to seven linc-genes, are specific solely for one cancer type (PAAD, STAD, HNSC, or LIHC), as shown in Table 3.

In groups with low and high expression of linc-genes, the calculated difference in survival rates reached at least a 2-fold value at a follow-up period of 24–48 months. Figure 4A demonstrates the Kaplan–Meier plot of the overall survival of patients with melanoma treated with immune checkpoint inhibitors (ICIs), which correlates with *LINC00305* gene expression levels. In addition, Figure 4B shows the post-progression survival of patients with gastric cancer, which correlates with *LINC01539* and *LINC01541* expression levels. At a follow-up period of 36 months, median survival rates were 21 and 8.5 months in low and high expression groups, respectively.

Immunotherapy is an effective option for treatment of malignant neoplasms. However, only a small portion of patients with cancer achieve positive responses to ICIs, mainly, to inhibitors of PD-1 (programmed cell death protein 1), PD-L1 (programmed death-ligand 1), and CTLA-4 (cytotoxic T-lymphocyte-associated protein 4). Therefore, the identification of predictive biomarkers of tumor susceptibility or resistance to ICIs will help to overcome this clinical complication. The ROC-plotter tool [66] was used to explore associations between expression levels of linc-genes and responses to ICIs. Downregulation of *LINC01415* in metastatic melanoma indicated responders to anti-PD-1 treatment, and the area under the curve value (AUC value) of the model and gene expression fold-change were equal to 0.725 and 2.0, respectively (Figure 5A,B). First, we selected the potential markers based on lincRNAs’ gene expression with a predictive value and model quality that correspond to AUC values > 0.7 (good quality). Second, regarding metastatic melanoma, *LINC00667* and *LINC00526* gene expression levels were also associated with responses to inhibitor of CTLA-4 (ipilimumab), but with AUC values < 0.7 (Appendix A) and a smaller number of clinical cases compared to *LINC01415*.

## 5. Pharmacologic Aspects of Genes Expression of lincRNAs

Using the PanDrugs2, adapted for personalized treatment selection of patients with cancer through analysis of gene–drug interactions [67], we found that deletion of *LINC02864* was associated with resistance to entinostat (a histone deacetylase inhibitor, clinical trials NCT01349959, NCT01038778, and NCT01305499) and AZD8186 (inhibitor of PI3Kβ- and δ-mediated AKT signaling, clinical trials NCT04001569 and NCT01884285). 

A search for transcriptomic signatures of chemotherapy response in the NCBI GEO repository using the ncRNADrug tool [68] (at |log_2_FC| ≥ 1 and FDR < 0.05) allowed us to find the following records. First, downregulation of *LINC00470*, *LINC00526*, and *LINC00667* genes in MCF-7 cell lines (breast cancer) was associated with sensitivity to doxorubicin. Second, upregulation of *LINC00667*, *LINC01416*, and *LINC01929* genes in LN229 and U87 cell lines (glioblastoma) was associated with resistance to temozolomide, while downregulation of *LINC00668* and *LINC00907* genes pointed to sensitivity to this drug.

The SigCom-Library tool [69] helps to provide a signature similarity search for mimickers and reversers, as well as gene–drug associations based on transcriptomic profiling data of cancer cell lines being exposed to various concentrations of drugs. Appendix A presents a summary of the anticancer activity of several candidate drugs as ‘reversers’ that were predicted by the transcriptomic signatures of eight linc-genes with differential expression (*LINC00305*, *LINC00526*, *LINC00667*, *LINC00668*, *LINC00907*, *LINC01254*, *LINC01443*, and *LINC01478*) in relation to breast and prostate cancers, as well as lymphoma and leukemia. These drugs were applied in nanomolar concentrations in cell-based assays. Among them, there are receptor tyrosine kinase inhibitors (ibrutinib, lapatinib, lucitanib, quizartinib, rebastinib, and tozasertib), histone deacetylase inhibitor (givinostat), and repurposing drugs (e.g., talinolol and tizanidine; Appendix A). Thus, we also demonstrated that the gene expression patterns of lincRNAs of human chromosome 18 might have relevance for prediction of anticancer drugs and immunotherapy responses.

## 6. Discussion

At least 12 lincRNAs of human chromosome 18 may play certain roles in the malignant transformation of cells, which are mediated via regulation of RNA–RNA and RNA–protein interactions. The main theses from 20 articles addressing these 12 lincRNAs are presented in Table 4. LincRNAs affect pro-tumorigenic processes, such as cell proliferation, migration, invasion, apoptosis, cell senescence, regulation of epithelial–mesenchymal transition, and angiogenesis, which can be applied to a small group of the most studied lincRNAs (*LINC00470*, *LINC00667*, *LINC00668*), while for the remaining ten lincRNAs mentioned in Table 4, investigations in the cancer field are still only occasional. Hence, the majority of lincRNAs of human chromosome 18 have been poorly studied not only in the pan-cancer context, but also in the basic functional aspect. We conducted additional analysis of transcriptomic, interactomic, and other available biomedical data on this group of lincRNAs, but excluding comparative analysis with any other groups of cancer-associated non-coding RNAs. To systematize the collected data on potential associations of each of the studied lincRNAs of human chromosome 18 with cancer, positive (+1) or negative (−1) values were assigned to 16 different hallmarks depending on their presence or absence, respectively. The clustered heat map of hallmarks’ distribution, including a set of 47 lincRNAs of human chromosome 18, is presented in Appendix A. No relevant data were found for the remaining 18 lincRNAs. Among 47 lincRNAs, the first cluster, shown in the far-left part of Appendix A, can be distinguished. This cluster consists of *LINC00470*, *LINC00667*, and *LINC00668*, as well as *LINC00305* and *LINC00526*, which have positive values of most hallmarks, except for hallmarks ‘A’ (somatic gene mutations), ‘B’ (gene fusions), and ‘C’ (gene knockouts effects). Therefore, based on the literature evidence and biomedical data mining, these five lincRNAs are cancer-associated and seem to be directly involved in malignant transformation of cells. The second cluster, shown in the far-right part of Appendix A, is represented by eleven lincRNAs (*LINC00907*, *LINC01254*, *LINC01387*, *LINC01415*, *LINC01416*, *LINC01443*, *LINC01477*, *LINC01478*, *LINC01538*, *LINC01539*, and *LINC01544*), which are characterized by hallmarks ‘D’ (copy number variations), ‘G’ (cancer-type-specific differential expression), ‘M’ (predicted relations in cancer-associated pathways), ‘O’ (potential prognostic value), and ‘Q’ (gene expression signatures for drug prediction). This cluster of lincRNAs is characterized by mediocre or hypothetical data on their involvement in cancer-associated processes, so they can be considered as molecular entities moderately or poorly studied in the pan-cancer context. The third cluster (the central part of Appendix A) includes 31 lincRNAs that also form many smaller sub-clusters, pointing to the significant heterogeneity of existing data, which does not yet allow us to hypothesize about their associations with cancer (understudied lincRNAs). The most common hallmarks for all three clusters of lincRNAs are ‘D’ (copy number variations), ‘F’ (tissue-specific expression), and ‘O’ (potential prognostic value).

Although this review was primarily focused on synthesizing literature data and the current state-of-the-art of genomic, transcriptomic, epigenomic, and interactomic data on 65 lincRNAs of human chromosome 18, we also touched on some structural aspects. Using data on nucleotide sequences presented in Appendix A, we predicted secondary structures for all 65 lincRNAs (Appendix A). All sequences were also pairwise-aligned, and the matrix of sequence identity of lincRNAs is shown in Appendix A. Only eight pairs of lincRNAs demonstrated high overall sequence identity (≥70%). Among them, we selected four pairs (*LINC01916*/*LINC01415*, *LINC02564*/*LINC01919*, *LINC02879*/*LINC01926*, and *LINC02564*/*LINC01925*) having similar secondary structure motifs (*p*-values < 0.1). Therefore, we tried to predict protein interactors for such motifs (Appendix A). The common protein interactors found may indicate similar functions of each pair of compared lincRNAs. This complements our findings on a pool of common protein interactors of lincRNAs of human chromosome 18 (Table 2), as well as proteins with potential to interact with small protein open-reading frames (smORFs) encoded by some lincRNAs (Appendix A).

## 7. Conclusions

The group of long intergenic non-coding RNAs (lincRNAs) of human chromosome 18 are poorly characterized molecular entities both in functional and disease-associated contexts. We have systematized the up-to-date literature data indicating the emerging functional roles of lincRNAs in cancer biology, but this applies to a relatively small subgroup of lincRNAs of human chromosome 18. However, for most lincRNAs, there is a significant gap in the understanding of their contribution to the molecular pathogenesis of diseases, in particular, widely spread solid cancers and malignant proliferative diseases. Therefore, we conducted a search for biomedical data and systems biologic analysis to create a panoramic view, mainly focusing on gene expression and interactomic data for the whole group of lincRNAs of human chromosome 18, which allowed us to consider some lincRNAs as potential candidates for future cancer investigations.

## Figures and Tables

**Figure 1 biomedicines-12-00544-f001:**
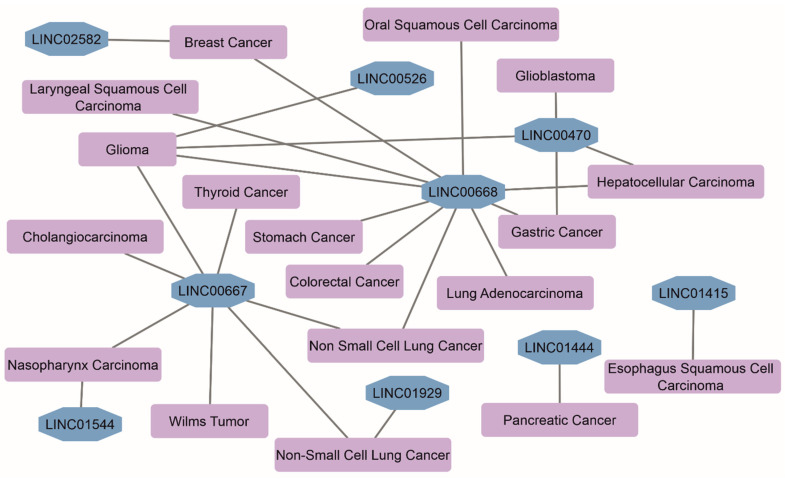
The network of associations between diseases and lincRNAs of chromosome 18 (data were retrieved from the RNADisease v. 4.0 repository database [26]; selection of disease terms was performed based on a score > 0.9 and experimental evidence).

**Figure 2 biomedicines-12-00544-f002:**
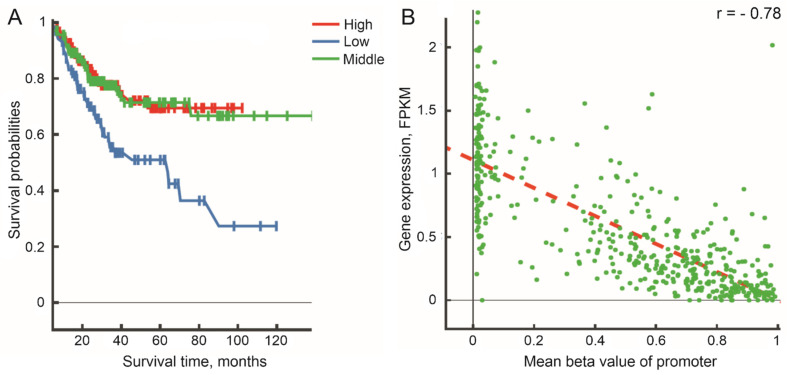
(**A**) Associations of promoter methylation of *LINC00526* gene with progression-free survival (*p*-value log-rank test = 6.02 × 10^−4^) in TCGA_UCEC cohort. (**B**) A plot of gene expression levels vs. mean beta values of promoter methylation (Spearman correlation coefficient r = −0.78, *p*-value = 2.46 × 10^−95^) in TCGA_UCEC cohort. A regression line is indicated by a red dashed line. DNA methylation beta values of zero–0.3, 0.3–0.7, and 0.7–1 were divided into low-, middle-, and high-risk groups of patients, respectively. FPKM—fragments per kilobase million (RNA-seq data). Images are the original outputs from the web-based tool DNA Methylation Interactive Visualization Database (DNMIVD).

**Figure 3 biomedicines-12-00544-f003:**
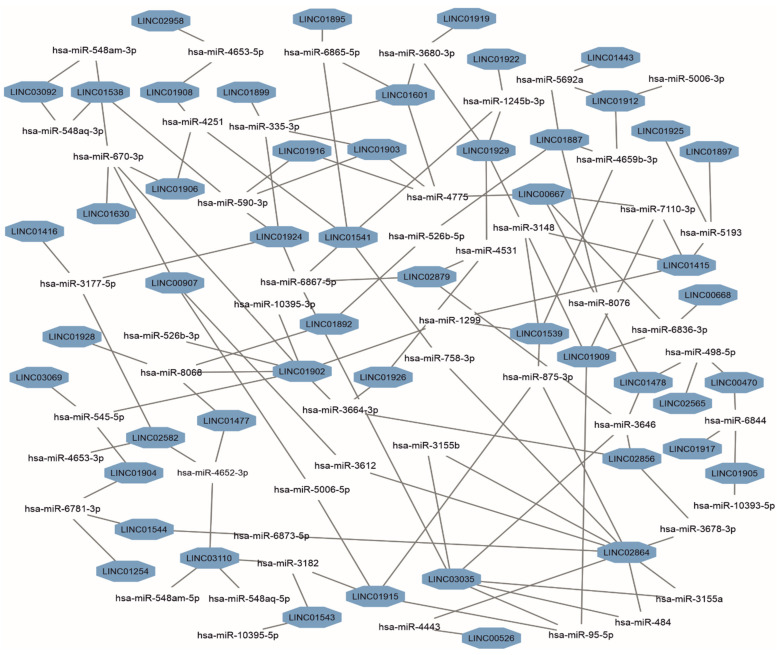
A subnetwork of predicted miRNAs interacting with three or more lincRNAs of human chromosome 18. The subnetwork was generated based on data retrieved from the miRDB portal.

**Figure 4 biomedicines-12-00544-f004:**
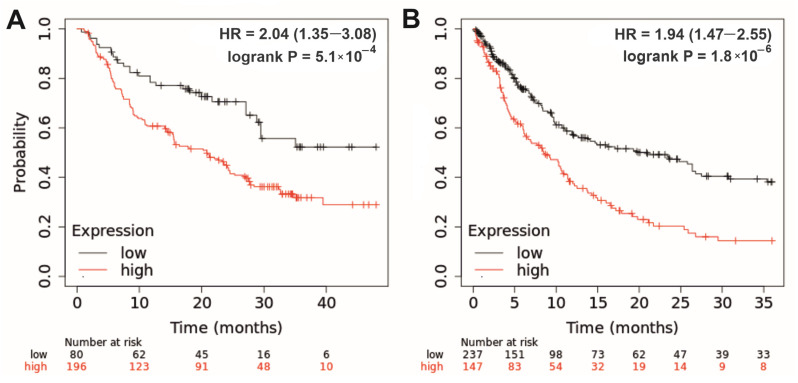
(**A**) Overall survival of patients with melanoma correlating with *LINC00305* gene expression: restrict analysis to ‘anti-PD-1, anti-PD-L1, anti-CTLA-4 treatment’. Upper quartile survivals were 18.1 and 7.2 months in low and high expression cohorts, respectively. Follow-up threshold = 48 months; FDR = 5%. (**B**) Post-progression survival of patients with gastric cancer correlating with *LINC01539* and *LINC01541* gene expression. Median survival rates were 21 and 8.5 months in low and high expression cohorts, respectively. Follow-up threshold = 36 months; FDR = 1%. HR (hazard ratio) is indicated for the high expression group. Kaplan–Meier analysis was performed using the KMplotter web-based tool [64,65].

**Figure 5 biomedicines-12-00544-f005:**
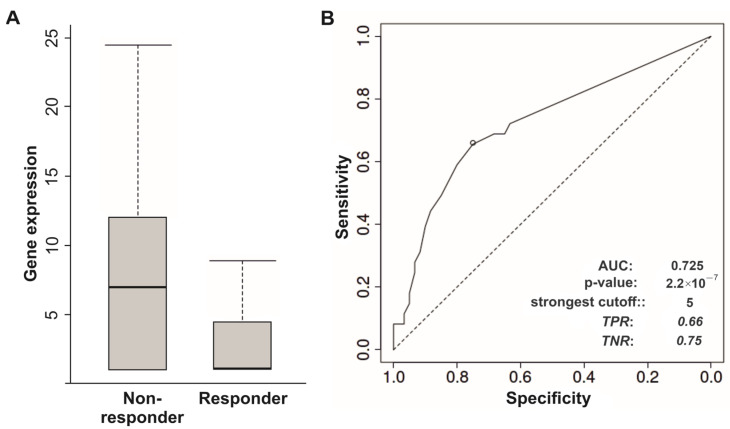
Association of *LINC01415* gene expression levels in metastatic melanoma (81 responders and 105 non-responders) with response to anti-PD-1 therapy. (**A**) Gene expression levels of *LINC01415*. (**B**) Sensitivity and specificity of the model. Mann–Whitney test *p*-value = <0.001, fold-change = 2. Analysis was performed using the web-based tool ROC-plotter [66].

**Table 1 biomedicines-12-00544-t001:** Differential expression of lincRNAs and microRNAs in tumor tissues.

Changes in Gene Expression (log_2_FC) *	Tumor Tissue	microRNA Expression (log_2_FC) **
*LINC00667* (2.9), (↓)	UCEC	miR-429-3p (5.3), (↑); miR-34a-5p (1.5), (↑); miR-877-5p (3.7), (↑); miR-142-5p (2.3), (↑); miR-183-5p (6.6), (↑); miR-106a-5p (3.4), (↑); miR-19a-3p (2.0), (↑); miR-3934-5p (2.8), (↑); miR-93-5p (2.4), (↑); miR-106b-5p (2.2), (↑); miR-17-5p (2.4), (↑); miR-454-3p (1.5), (↑); miR-3016-3p (3.2), (↑)
*LINC00668* (3.1), (↑)	STAD	miR-204-5p (2.0), (↓)
*LINC00668* (1.6), (↑)	READ	miR-236-3p (1.2), (↓)
*LINC01539* (1.8), (↓)	THCA	miR-34a-5p (2.0), (↑)

* log_2_FC (log_2_fold-change) in tumor/normal tissues (analysis of TCGA pan-cancer cohort was performed using the web-based tool GEPIA2 [32], *p*-value < 0.01). ** log_2_FC microRNAs’ expression in tumor/normal tissues (analysis of TCGA pan-cancer cohorts was performed using the web-based tool MIR-TV [45], *p*-value < 0.01). Abbreviations: UCEC—uterine corpus endometrial carcinoma; STAD—stomach adenocarcinoma; READ—rectal adenocarcinoma; THCA—thyroid cancer; (↓) and (↑) represent down- and up-regulation, respectively.

**Table 2 biomedicines-12-00544-t002:** Functional enrichment analysis of cellular proteins interacting with lincRNAs of human chromosome 18.

Functional Terms, ID *	Proteins
mRNA surveillance pathway, hsa03015	CSTF2T, NCBP2, UPF1, WDR33
Metabolism of RNA, R-HSA-8953854	CSTF2T, FBL, FUS, HNRNPU, IGF2BP1, IGF2BP3, LSM11, NCBP2, POLR2A, PRPF8, U2AF1, UPF1, WDR33
mRNA processing, WP411	CSTF2T, FUS, HNRNPU, NCBP2, POLR2A, PRPF8, U2AF1
Regulation of gene expression, epigenetic, GO:0040029	AGO1, CTCF, ESR1, HNRNPU, MOV10, POLR2A, POU5F1, UPF1
RNA splicing, GO:0008380	CSTF2T, FUS, HNRNPU, NCBP2, POLR2A, PPIG, PRPF8, QKI, RBM10, TIA1, WDR33
Post-transcriptional regulation of gene expression, GO:0010608	AGO1, DDX3X, DHX36, ESR1, HNRNPU, IGF2BP1, IGF2BP3, MOV10, NCBP2, POLR2A, POU5F1, QKI, RBM10, TIA1, UPF1
Epithelial cancers, PA447242	AR, ESR1, EWSR1, FOXA1, IGF2BP3, POU5F1, SOX2
Urogenital neoplasms, PA445995	AR, ESR1, FOXA1, IGF2BP3, POU5F1, SMARCA4, SOX2
Adenocarcinoma, PA443265	ESR1, FOXA1, HNF4A, IGF2BP3, POU5F1, SOX2

* WebGestalt web-based tool [50] was used for functional enrichment analysis at the following settings: reference gene list—‘genome protein-coding’; minimum number of genes for a category—‘3’; multiple test adjustment—‘Benjamini–Hochberg’; significance level—‘FDR < 0.05’; redundancy reduction—‘weighted set cover’.

**Table 3 biomedicines-12-00544-t003:** Transcriptomic signatures of lincRNAs of human chromosome 18 with potential prognostic significance.

RNA-Seq Data			
Genes	Cancer Type	Survival	Description
*LINC01443*, *LINC01538*, *LINC01910*, *LINC01916*, *LINC01925*, *LINC01929*	PAAD *	OS **	Upper quartile survival: low and high expression cohorts—19.73 and 9.27 months, respectively; HR *** = 2.45 (1.51–4), *p*-value = 0.0002; FDR = 2%; follow-up threshold = 24 months. Cancer specificity: absolute
*LINC01416*, *LINC01544*, *LINC01900*, *LINC01910*, *LINC01922*, *LINC01926*, *LINC02565*	STAD	PFS	Median survival: low and high expression cohorts—(no data); HR = 3.47 (1.72–6.98), *p*-value = 0.0002; FDR = 2%; follow-up threshold = 24 months. Cancer specificity: absolute
*LINC01443*, *LINC01478*, *LINC01900*, *LINC01899*, *LINC01538*, *LINC01541*	HNSC	PFS	Median survival: low and high expression cohorts—(no data); HR = 0.2 (0.09–0.48), *p*-value = 0.00004; FDR = 1%; follow-up threshold = 24 months. Cancer specificity: absolute
*LINC00305*, *LINC01254*, *LINC01387*, *LINC01477*	LIHC	OS	Upper quartile survival: low and high expression cohorts—11.47 and 23.7 months, respectively; HR = 0.48 (0.33–0.7), *p*-value = 0.00008; FDR = 1%; follow-up threshold = 48 months. Cancer specificity: absolute
*LINC01478*, *LINC01538*, *LINC01539*, *LINC01541*	LIHC	OS	Upper quartile survival: low and high expression cohorts—12.17 and 25.6 months, respectively; HR = 0.44 (0.3–0.65), *p*-value = 0.00005; FDR = 1%; follow-up threshold = 36 months. Cancer specificity: absolute
Gene chip data			
*LINC00470*, *LINC00907*, *LINC01477*	COAD	PPS	Median survival: low and high expression cohorts—(no data); HR = 5.87 (1.97–17.5), *p*-value = 0.00032; FDR = 2%; follow-up threshold = 24 months.
*LINC01539*, *LINC01541*	STAD	PPS	Median survival: low and high expression cohorts—21 and 8.5 months, respectively; HR = 1.94 (1.47–2.55), *p*-value = 0.0000018; FDR = 1%; follow-up threshold = 36 months.

* COAD—colon adenocarcinoma; HNSC—head and neck squamous cell carcinoma; LIHC—liver hepatocellular carcinoma; PAAD—pancreatic adenocarcinoma; STAD—stomach adenocarcinoma. ** OS—overall survival; PFS—progression-free survival; PPS—post-progression survival; FDR—false discovery rate. *** HR—hazard ratio (high expression group). Survival rates were calculated using the KMplotter web-based tool [64,65].

**Table 4 biomedicines-12-00544-t004:** Literature evidence on participation of lincRNAs of chromosome 18 in cancer-associated processes.

Genes	Description	References
*LINC00305*	*LINC00305* exhibits an upregulated expression in gastric cancer and regulates the Wnt/β-catenin signaling pathway to promote cell proliferation and inhibit apoptosis.	[70]
*LINC00470*	*LINC00470* is associated with PTEN mRNA and suppresses its stability through interaction with methyltransferase 3.	[71]
	*LINC00470* directly interacts with FUS RNA-binding protein, serving as an AKT activator to promote glioblastoma progression.	[72]
	Functional studies show that knockdown of *LINC00470* expression inhibits hepatocellular carcinoma cell proliferation and cell cycle progression, while overexpression of *LINC00470* shows the opposite effects.	[73]
*LINC00526*	Silencing the expression of *LINC00526* inhibits glioma cell growth and invasion. *LINC00526* functions as a sponge for miR-5581-3p to regulate brain-expressed X-linked 1 (BEX1) expression.	[74]
*LINC00667*	*LINC00667* knockdown significantly inhibits colorectal cancer cell growth and migration. YY1 transcription factor induces the upregulation of *LINC00667*, and miR-449b-5p interacts with *LINC00667*.	[75]
	*LINC00667*, acting as a tumor promoter, recruits eukaryotic translation initiation factor 4A3 (EIF4A3) to stabilize vascular endothelial growth factor A (VEGFA) mRNA for modulation of non-small cell lung cancer (NSCLC) progression.	[76]
	*LINC00667* plays a critical role in metastatic esophageal cancer by mediating the sponge regulatory axis miR-200b-3p/SLC2A3 (glucose transporter).	[77]
	*LINC00667* is a molecular sponge in the miR-130s-3p/AR (androgen receptor) signal pathway in the progression of hepatocellular carcinoma, in which it relieves the repressive function of miR-130a-3p on the *AR* expression.	[78]
*LINC00668*	*LINC00668* is highly expressed in breast cancer (BC) tissues and can promote the progression of BC by inhibiting apoptosis and accelerating cell cycle progression.	[79]
	Transcription factor E2F1-activated *LINC00668* enriches the mechanistic link between lncRNA and the E2F1-mediated cell cycle regulation pathway and may serve as a target for new therapies in gastric cancer.	[80]
	STAT3-induced *LINC00668* contributes to NSCLC progression through upregulating Krüppel-Like Factor 7 (KLF7) expression by sponging miR-193a and may serve as a potential target for therapies.	[81]
*LINC01255*	*LINC01255* can interact with proto-oncogene BMI1 and repress the transcription of MCP-1 (C-C motif chemokine ligand 2) to active the p53–p21 pathway, thus inhibiting the senescence of human mesenchymal stromal cells and proliferation of acute myeloid leukemia cells.	[82]
*LINC01541*	Estradiol promotes the synthesis of VEGFA by altering the expression levels of *LINC01541* and miR-429, thereby affecting the angiogenesis process of endometrioid adenocarcinoma.	[83]
*LINC01895*	*LINC01895* overexpression enhances cell proliferation, migration, and invasion, and inhibition of cell apoptosis. Results obtained illustrate the role of *LINC01895* in cisplatin resistance of lung adenocarcinoma, suggesting the potential of *LINC01895* as a new therapeutic target.	[84]
*LINC01915*	*LINC01915* inhibits the normal fibroblast uptake of colorectal cancer-derived extracellular vesicles via the miR-92a-3p/KLF4/CH25H axis, thus preventing tumor growth.	[85]
*LINC01924*	For BK polyomavirus, there are three consensus integration sites between the primary and metastatic tumors, which affect *LINC01924*.	[86]
*LINC01929*	Overexpression of *LINC01929* promotes bladder cancer development, while overexpression of miR-6785-5p inhibits bladder cancer development. *LINC01929* acts as a sponge for miR-6785-5p and partially reverses the role of miR-6785-5p.	[87]
	*LINC01929* is upregulated in NSCLC tissues and cell lines and associated with later clinical stages. Downregulation of *LINC01929* inhibits cellular proliferation, migration, and invasion by targeting miR-1179.	[88]
*LINC02582*	LINC02582 is a downstream target of miR-200c linking miR-200c to CHK1, and miR-200c increases radio-sensitivity by downregulation of CHK1. Overexpression of *LINC02582* promotes radio-resistance of cancer cells.	[89]

## Data Availability

Predicted secondary structures of lincRNAs of human chromosome 18 (json format) are deposited in FigShare, https://doi.org/10.6084/m9.figshare.25257343 (accessed on 21 February 2024).

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
