# Peer review of "Long Intergenic Non-Coding RNAs of Human Chromosome 18: Focus on Cancers"

_biomedicines, 2024, doi:10.3390/biomedicines12030544_

Round 1
Reviewer 1 Report
Comments and Suggestions for Authors
Some minor adjustments are needed before publishing the manuscript:
Major comments:
1) The entire work lacks a clear description of key findings. The concept that lincRNAs are important for cancer is somewhat vague. After reading the entire work, it's challenging to pinpoint the most crucial findings. For instance, in the second part of the abstract, authors should summarize their key findings instead of listing all tools and databases used.
2) The primary hypothesis and focus of this work should demonstrate that lincRNAs on chromosome 18 are crucial for cancer (this is well done). However, it is not clear whether these lincRNAs are overrepresented in this aspect compared to other RNAs (lncRNAs, microRNAs) or proteins. It can be expected that any group of proteins or RNAs will contain some representatives connected to cancer, but it does not imply that the whole group (here lincRNAs) should be treated as a marker of cancer.
3) The data related to the analysis should be stored in bioinformatic (e.g., fasta) or general (e.g., csv, json) formats:
- At the very least, provide all 65 lincRNA sequences as a separate fasta file.
- Store all results for analyses mentioned in figures and tables (if needed), e.g., data behind Figure 1, Figure 4, the first part of the 2.3 paragraph, etc.
4) Extend the analysis by predicting the secondary structure of all 65 lincRNAs and try to analyze the results in this respect too.
The requested data can be compressed into one repository and added as supplementary data directly to the manuscript or included in external repositories such as Zenodo, Dryad, Figshare, or Open Science Framework.
Minor:
- Line 70 - In Table S1, mention that some lincRNAs are associated with membranes (not listed in the text).
- Lines 71-72: "frames uncharacterized" --> "frames of uncharacterized"
- Line 83: "thr" --> "the"
- Table S3 should be a parsable table with actual numbers that can be copied, instead of a flattened image (screenshot) depicting the table.
- Figure 2 has some cropping problems (some parts look like poorly erased labels in Paint). In panel B, add the value of correlation in the right-upper corner.
- All plots in Figure 3 and Figure S1 should use the same y-axis height (range). Additionally, improve the resolution, and make labels under the x-axis larger.
- Figure 4 (and, in general, all figures) should mention the database/method used, here the miRDB database.
- Figure S2 should be a table. Additionally, in its current form, the black background is distracting.
- Lines 322-323: "Nie Q. and co-authors" --> "Nie and co-authors"
- Table 4 - Use the same style of borders as in previous tables.
- Line 369: "data not shown" - Delete this and add the data (there is no limit in the supplement, and this is just a small part of what you already added).
- Fig. 6 - Add A and B labels to panels. Additionally, the left panel is too big/wide.
see above
Author Response
The authors would like to express their gratitude to the reviewer for providing very valuable comments and for their scrupulous, albeit critical, review, which has greatly contributed to improving the current work.
The major corrections:
- The abstract section has been rewritten according to the reviewer’s recommendations.
- The supplementary material has been added according to the reviewer’s recommendations (now it includes 24 Tables and 7 figures).
- The supplementary material has been added with description of some structural features of lincRNAs of human chromosome 18, including file with sequences, predicted secondary structures, multiple alignments, etc. (according to the reviewer’s recommendations).
- All reviewer’s minor and major comments have been addressed.
- The section ‘Discussion’ has been added in the text.
- The section ‘Conclusion’ has been rewritten.
- We have added seven new references.
- Some appropriate corrections in the main text have been done and highlighted with bold font and yellow color.
Major comments
Reviewer
1) The entire work lacks a clear description of key findings. The concept that lincRNAs are important for cancer is somewhat vague. After reading the entire work, it's challenging to pinpoint the most crucial findings. For instance, in the second part of the abstract, authors should summarize their key findings instead of listing all tools and databases used.
Authors
Agree.
Corrected accordingly.
Key findings are described in the Abstract section.
Reviewer
2) The primary hypothesis and focus of this work should demonstrate that lincRNAs on chromosome 18 are crucial for cancer (this is well done).
However, it is not clear whether these lincRNAs are overrepresented in this aspect compared to other RNAs (lncRNAs, microRNAs) or proteins.
It can be expected that any group of proteins or RNAs will contain some representatives connected to cancer, but it does not imply that the whole group (here lincRNAs) should be treated as a marker of cancer.
Authors
We agree that it is not quite correct to draw a conclusion about the involvement of the entire target group of lincRNAs in cancer based on the bioinformatics analysis performed. In the ‘Discussion’ section, the findings on lincRNAs of human chromosome 18 are summarized and graphical view is added in the Figure S7. This allowed us to select subgroups (clusters) of lincRNAs of human chromosome 18 and further considered them in terms of associations with cancer by different characteristics.
Below, we provide some explanations of the concepts and methods used in the review.
First, we examined publicly available transcriptomics data to search for connections between the expression of lincRNAs of human chromosome 18 and clinical outcomes (survival rates) as well as responses to therapy (Section 4). We described the finding on the potential prognostic value of transcriptomics signatures containing two and more lincRNAs of chromosome 18 and potential predictive value of individual lincRNAs.
Second, we did not consider the found connections of lincRNAs of human chromosome 18 with other miRNAs and proteins as indirect evidence in favor of any biomarker value of target lincRNAs. They were considered in terms of their associations with molecular carcinogenesis and their potential involvement in cancer through the interactions with microRNAs and proteins. Some of these have already been identified as cancer-associated molecular entities.
Third, as part of the analytical review, we did not aim to perform a sufficiently large comparative analysis of the target group of lincRNAs of human chromosome 18 with other groups of non-coding lncRNAs (for example, lincRNAs of human chromosome 19) in the pan-cancer context. The aim of the review was to collect, analyze and systematize data related to only 65 lincRNAs of human chromosome 18, that is, to conduct a chromosome-centric analysis to characterize each of lincRNAs as a molecular object in terms of possible future research.
Reviewer
3) The data related to the analysis should be stored in bioinformatic (e.g., fasta) or general (e.g., csv, json) formats:
- At the very least, provide all 65 lincRNA sequences as a separate fasta file.
- Store all results for analyses mentioned in figures and tables (if needed), e.g., data behind Figure 1, Figure 4, the first part of the 2.3 paragraph, etc.
Authors
Agree. Corrected accordingly.
We have added all the available auxiliary data (Tables, Figures, gene sets) used in the Supplementary materials.
Reviewer
4) Extend the analysis by predicting the secondary structure of all 65 lincRNAs and try to analyze the results in this respect too. The requested data can be compressed into one repository and added as supplementary data directly to the manuscript or included in external repositories such as Zenodo, Dryad, Figshare, or Open Science Framework.
Authors
The supplementary material has been added with files containing sequences, predicted secondary structures, multiple alignments, etc. (according to the reviewer’s recommendation).
As we have found earlier, most lincRNAs belong to the relatively poorly annotated group in terms of their general functional and disease-specific contexts. Therefore, in addition to literature mining, we used biomedical data mining, mainly, by analysis of transcriptomic and interactomic data to find / predict associations of lincRNAs with cancer. It was a basic search method in our analytical review, while analysis of structural features of target lincRNAs was not a priority. One of the most common (classical) methods for annotating poorly/fragmentarily studied molecular entities, such as lincRNAs of human chromosome 18, is an examination of the spectrum of interactors (RNAs and proteins) to perform the annotation based on the known functions of these interactors. For example, linc2function (https://bioinformaticslab.erc.monash.edu/linc2function) adapted for the analysis of non-coding RNA sequences includes such a section for gene annotation. We suppose that, at present, the prediction of functions or disease associations of non-coding RNAs based on analysis of their primary sequence or secondary structure features is quite a challenge required the specific software products based on machine learning algorithms.
Minor comments:
Reviewer
- Line 70 - In Table S1, mention that some lincRNAs are associated with membranes (not listed in the text).
Authors
- Added in the text
Reviewer
- Lines 71-72: "frames uncharacterized" --> "frames of uncharacterized"
Authors
- Added ‘of’ in the text
Reviewer
- Line 83: "thr" --> "the"
Authors
- Added ‘the’ in the text
Reviewer
- Table S3 should be a parsable table with actual numbers that can be copied, instead of a flattened image (screenshot) depicting the table.
Authors
- Corrected accordingly
Reviewer
- Figure 2 has some cropping problems (some parts look like poorly erased labels in Paint).
Authors
- We used the DNA Methylation Interactive Visualization Database (DNMIVD) (http://www.unimd.org/dnmivd/) [PMID: 31598709] for analysis the promoter methylation of genes, and Figure 2A and 2B are the original outputs. A caption for Figure 2 was added ‘Figures are the original outputs from the web-based tool DNA Methylation Interactive Visualization Database (DNMIVD)‘.
Reviewer
- In panel B, add the value of correlation in the right-upper corner.
Authors
- Corrected accordingly
Reviewer
- All plots in Figure 3 and Figure S1 should use the same y-axis height (range). Additionally, improve the resolution, and make labels under the x-axis larger.
Authors
- Corrected accordingly, where applicable.
Figure 3 was deleted from main text and pasted to the Supplementary material.
Reviewer
- Figure 4 (and, in general, all figures) should mention the database/method used, here the miRDB database.
Authors
- Corrected accordingly.
Reviewer
- Figure S2 should be a table. Additionally, in its current form, the black background is distracting.
Authors
- We added the corrected table instead of distracting Figure S2 in the Supplementary materials.
Reviewer
- Lines 322-323: "Nie Q. and co-authors" --> "Nie and co-authors"
Authors
- Corrected accordingly.
Reviewer
- Table 4 - Use the same style of borders as in previous tables.
Authors
- Corrected accordingly.
Reviewer
- Line 369: "data not shown" - Delete this and add the data (there is no limit in the supplement, and this is just a small part of what you already added).
Authors
- In this case, the predictive models with AUC-values less than 0.7 were not considered in detail, since the quality of these models is quite mediocre. We added in Figure S6 the finding regarding metastatic melanoma, but with less favorable model’s parameters compared to the model shown in Figure 5 (the main text). Corrected accordingly.
Reviewer
- Fig. 6 - Add A and B labels to panels. Additionally, the left panel is too big/wide.
Authors
- Corrected accordingly.

Reviewer 2 Report
Comments and Suggestions for Authors
Various, diverse non-coding RNAs play critical roles in cancer development and progression. Human chromosome 18 is an emerging but largely under-appreciated "hotspot" for lincRNAs regulating neoplastic cellular transformation via molecular interactions with microRNAs and proteins. This review focuses on genomic, transcriptomic, epigenomic, interactomic, and literature data on human chromosome 18's lincRNAs in cancer biology and provides an up-to-date analytical review in this regard. Web-based tools and databases, including BioGRID, cBioPortal, DepMap, DNMIVD, GeneCards, GEO, GEPIA2, KM-plotter, LncBook, LncTarD, miEAA, miRDB, NCpath, NPInter, PepPsy, PlasmiR, RBPmap, RNAlocate, RNAinter, ROC-plotter, SmProt, SigCom-LINCS, Webgestalt, are used for bioinformatic analysis. Overall, this Review provides a multi-omics examination and analysis of lincRNAs of chromosome 18 as related to their tumorigenic potential by analyzing gene alterations, expression patterns, RNA-RNA and RNA-protein interactions. Finally, the prognostic and potential pharmacological value of lincRNAs in common cancer types is explored.
Overall, this is a focused but thorough review. Figures are appropriate and useful. This Review will be of interest and value to the field.
Comments on the Quality of English LanguageWell-written, no significant issues noted
Author Response
We greatly thank the reviewer for positive assessment of our review and interest in the research of poorly studied macromolecules contributing to human diseases.
Round 2
Reviewer 1 Report
Comments and Suggestions for Authors
The authors addressed my concerns, thus the manuscript can be published after addressing minor issues:
1) Please double-check and correct small editorial errors (e.g., l.532 "At least, At least," - it seems like a minor error, perhaps this occurred due to an uncleaned version with yellow marks. Nevertheless, these appear to be small issues that can even be resolved during proofreading).
2) The previous Figure 3 was split into multiple small supplementary figures (this is the authors' choice, but I would prefer to stay with one multipanel figure normalized to the same scale to observe differences in strength in such a way, even if it would flatten some of the boxplots). Nevertheless, t can remain as it is now.
3) Whenever you use the xlsx format, please consider a less error-prone and machine-learning-friendly format in the future (e.g., csv, json).
4) For secondary structure (Table S22), consider adding raw files (in bioinformatics format) - the pictures are nice for the eye, but their resolution makes them completely unreadable at the nucleotide level, and even if they were readable, it is impossible to use/process such images in further analyses.
see above
Author Response
Reviewer
1) Please double-check and correct small editorial errors (e.g., l.532 "At least, At least," - it seems like a minor error, perhaps this occurred due to an uncleaned version with yellow marks. Nevertheless, these appear to be small issues that can even be resolved during proofreading).
Authors
We thank the reviewer for careful examination of the manuscript.
We have made some additional corrections.
Reviewer
2) The previous Figure 3 was split into multiple small supplementary figures (this is the authors' choice, but I would prefer to stay with one multipanel figure normalized to the same scale to observe differences in strength in such a way, even if it would flatten some of the boxplots). Nevertheless, t can remain as it is now.
Authors
We would like to keep this figure as it is, because we believe the current version is better for visual presentation.
Reviewer
3) Whenever you use the xlsx format, please consider a less error-prone and machine-learning-friendly format in the future (e.g., csv, json).
Authors
We thank the reviewer for the valuable recommendation.
Reviewer
4) For secondary structure (Table S22), consider adding raw files (in bioinformatics format) - the pictures are nice for the eye, but their resolution makes them completely unreadable at the nucleotide level, and even if they were readable, it is impossible to use/process such images in further analyses.
Authors
Agree. Predicted secondary structures (json format) are deposited in FigShare – doi:10.6084/m9.figshare.25257343.
